# Gamblers’ Perceptions of the Impact of the COVID-19 Pandemic on Their Gambling Behaviours: Analysis of Free-Text Responses Collected through a Cross-Sectional Online Survey

**DOI:** 10.3390/ijerph192416603

**Published:** 2022-12-10

**Authors:** Marianne Renard, Sophie Audette-Chapdelaine, Annie-Claude Savard, Sylvia Kairouz, Magaly Brodeur

**Affiliations:** 1Department of Psychology, Université du Québec à Montréal, Montreal, QC H2X 3P2, Canada; 2Department of Family Medicine and Emergency Medicine, Université de Sherbrooke, Sherbrooke, QC J1K 2R1, Canada; 3School of Social Work and Criminology, Université Laval, Quebec, QC G1V 0A6, Canada; 4Department of Sociology and Anthropology, Concordia University, Montreal, QC H3G 1M8, Canada

**Keywords:** gambling, COVID-19, qualitative, survey, pandemic

## Abstract

The COVID-19 pandemic has brought drastic changes to the lives of a substantial portion of the world’s population. Many stakeholders have expressed concern about the impact of the pandemic on gambling practices, which have historically increased during times of crisis. The purpose of this study was to provide a snapshot of the impact of the pandemic on gambling practices, focusing on the lived experiences of people who gamble. An online cross-sectional survey was conducted between 16 February and 15 March 2021. An open-ended question allowed the participants to describe in their own words the impact of the COVID-19 pandemic on their gambling practices. A qualitative analysis was conducted based on 724 responses to this question. Among the participants, 57% were problem gamblers, according to their Problem Gambling Severity Index score. Three themes were identified: (1) changes in gambling practices perceived by the respondents during the pandemic, (2) the impacts of these changes, and (3) the factors that influenced the changes in their gambling practices. A meaningful proportion of the sample of gamblers felt that their gambling practices had increased during the pandemic. Many of them did not report the deleterious effects of this increase, whereas others were devastated. Thus, variations in gambling practices during the pandemic must be interpreted with caution, as they may reflect a variety of realities.

## 1. Introduction

The COVID-19 pandemic resulted in drastic changes in the lives of a substantial portion of the world’s population (national lockdowns, social distancing, banned gatherings, telework, etc.) [1,2,3,4]. The gambling industry, as with many others, has been deeply shaken by the shutdown of many facilities around the world [5,6,7,8,9,10]. Gambling includes all activities that involve betting money (e.g., lotteries, scratch cards, slot machines, video lottery terminals, poker, online gambling, etc.), and it is estimated that between 65 and 82% of the world’s population engages in gambling activities each year [11,12]. About 1–3% of the general population are considered problem gamblers, meaning that their gambling habits result in negative consequences for themselves as well as their surroundings and community [13,14,15,16,17].

Since the beginning of the pandemic, many stakeholders worldwide (such as researchers, policymakers, and addiction workers) have expressed concern about the impact of the pandemic on people who gamble. Gambling practices tend to increase in crises [18,19,20,21], and problem gambling is positively correlated with symptoms of anxiety and depression [22,23], which have been on the rise since the beginning of the pandemic [24,25,26,27]. Moreover, stakeholders have predicted that the closure of land-based facilities would lead to a shift to online gambling [6,19,28], which is a high-risk activity [12,29,30] with a rate of problem gambling varying between 5% to 40% [31,32,33]. While the literature is still limited, there is a growing scientific conversation on the impact of the COVID-19 pandemic on gambling behaviors [34]. Certains consequences of the pandemic on populations, such as financial restrictions and monetary concerns, social isolation, as well as anxiety, stress, and boredom may have effects on gambling behaviors, either by reducing gambling activities or exacerbating problematic gambling [34,35,36,37]. In the province of Quebec, Canada, the pandemic has led to the closure of casinos, bingo halls, video lottery terminals, and horse racing, as well as to the suspension of the sale of lotteries at retail outlets [8]. A migration to online gambling has consequently been noted by stakeholders in the industry who are concerned about this situation [38].

Several studies have focused on the impacts of the COVID-19 pandemic on gambling practices [6,39,40,41,42]. However, few qualitative studies have been published on the subject. Qualitative studies on this topic are needed [6], as they bring an in-depth understanding of the experience of gamblers during the pandemic and could increase the interpretive value of the results obtained [43]. Qualitative analysis of responses to an open-ended question asked as part of a cross-sectional survey is a way to contextualize questionnaire responses, to draw out aspects of participants’ experiences that could not be identified from responses to closed-ended questions, and to obtain results that reflect participants’ priorities and perspectives rather than those of researchers. It is also a way to obtain a large amount of qualitative data about the experiences of a wide range of individuals [43,44].

The objective of this article is to draw up a portrait of the experiences of gamblers regarding the impact of the pandemic on their gambling practices based on the answers to an open-ended question included in a cross-sectional survey.

## 2. Materials and Methods

### 2.1. Study Design

The data of this qualitative descriptive study were drawn from a sequential explanatory mixed-methods study whose main objective was to draw a portrait of the impacts of the COVID-19 pandemic on gamblers in the province of Quebec (Canada) [45]. The first phase of the study consisted of an online cross-sectional survey on the impact of the pandemic on gamblers. This was conducted online from 16 February to 15 March 2021. The following optional single open-ended free-text question was placed at the end of the questionnaire, following the closed questions: “The COVID-19 pandemic has had a major impact on our lives. How would you describe the impact of the pandemic on your gambling practices? Please feel free to share any details you think are relevant. These will be helpful to us.” The responses to this question are the focus of this article.

### 2.2. Recruitment

Survey participants were recruited through non-randomized online sampling via social networks (Facebook) and the websites of partner organizations (Loto-Québec, Gambling: Help and Referral). Inclusion criteria were: (1) being aged 18 years and older, (2) living in the province of Quebec (Canada), and (3) having gambled at least once in the past year.

### 2.3. Questionnaire Design and Content

The 85-item online questionnaire was developed by the research team, helped by the patient-partner, and included validated questionnaires and tools. It was designed to provide a picture of the impact of COVID-19 on gamblers and to learn more about their history and co-morbidities as well as their experience of care. The questionnaire was divided into four parts: the socio-demographic profile, the general impact of COVID-19, the impact of COVID-19 on gambling practices, and the health profile and experience of care. It should be noted that the questionnaire and the responses of the participants were written in French. Consequently, the citations presented in this article are professionally revised translations.

The questionnaire measured, among other things, the Problem Gambling Severity Index (PGSI) [46], one of the tools used to assess the severity of gambling problems. Gamblers who score 0 on the Problem Gambling Severity Index (PGSI) are considered to be at no risk of developing gambling problems. Those who score between 1 and 2 are considered low risk, while those who score 3 or more are considered problem gamblers. Problem gamblers are divided into two categories: moderate risk gambling (3 ≤ PGSI ≤ 7) and likely pathological gambling (PGSI ≥ 8).

### 2.4. Ethics

The research protocol was approved by the Research Ethics Board of the Centre intégré universitaire de santé et de services sociaux—Centre hospitalier universitaire de Sherbrooke (CIUSSS de l’Estrie—CHUS). The data were anonymized, and all participants gave their written consent at the beginning of the online survey.

### 2.5. Analysis

The analysis of the responses to the open-ended free-text question was aimed at providing a snapshot of the participants’ experiences, expressed in their own words, as well as their most pressing concerns. An iterative-inductive methodology was employed throughout to capture emerging and recurring themes in the participants’ discourses. During an initial immersive reading, the principal author (MR) developed the first version of the thematic framework using an inductive-iterative method. This was then revised by two co-authors (MB and SAC). A systematic thematic analysis was then conducted by MR. During this phase, the thematic framework was enriched and refined. This version of the thematic framework was also revised and discussed with SAC and MB. Finally, the developed framework was used to systematically code the data set using NVivo software [38]. The analysis was carried out in French, translated, and professionally revised for the purpose of reporting.

## 3. Results

A total of 1529 people participated in the study. Of these, 969 completed the questionnaire to the end, and 724 responded to the optional open-ended questions analyzed in this article. Table 1 presents the socio-demographic portrait of the participants who answered the open-ended question. The median age of the participants was 43 years, and the sample was 54% female. Respondents were fairly evenly distributed across educational levels, except for the “no diploma” category, which is underrepresented. Most respondents are employees (58%). Lastly, we observed that 57% of the participants were problem gamblers based on their PGSI score.

Responses to the open-ended question varied in length and depth from one participant to the next, as would be expected with this type of question. Three main themes emerged from the analysis. The first reflects participants’ perceptions of changes in their gambling practices during the pandemic. This theme has three sub-themes. The second theme reflects the impact of these changes, as perceived by the participants. It has two sub-themes. The third theme reflects the factors perceived by participants to have influenced changes in their gambling practices during the pandemic. This theme has four sub-themes. All themes and subthemes are presented in Table 2, which also specifies the number of occurrences of each theme. When a theme or sub-theme was mentioned by a substantial proportion of participants, some weight was assigned to it in the analysis, while considering that because a participant did not mention a particular theme does not mean it did not apply to them. All responses could be associated with more than one theme or sub-theme.

### 3.1. Perceived Changes in Gambling Practices during the Pandemic

This theme presents the participants’ perceptions of changes in their gambling practices during the pandemic. It includes the following three sub-themes: perceived increase in gambling practices, perceived decrease in gambling practices, and perceived stability in gambling practices.

#### 3.1.1. Perceived Increase in Gambling Practices

Many participants felt that the pandemic had led to an increase in their gambling practices (n = 238). Most of these participants were regular gamblers before the pandemic, and many did not report any negative effects of their increased gambling during the pandemic. In some cases, they expressed that their increase in gambling seemed reasonable to them:


*I played a little more but kept control.*
(Participant 307, M, age 51, PGSI score 0)

However, many others simply did not provide any information regarding whether the increase in their gambling practices had deleterious effects:

*Having more [free] time, there was an increase in gambling time*.(Participant 5, F, age 61, PGSI score 12)

Some players expressed the negative impacts of their increased gambling, which are sometimes noteworthy:


*The pandemic did not create my problem; it simplified it drastically. It has made it completely free to exist without any barriers. Except the problem is that my life is suffering. I’m in danger of losing my home, and I’m embarrassed and too far gone to stop. It’s unhealthy and I can’t be the only one.*
(Participant 156, F, age 31, PGSI score 19)

Among these desperate gamblers, a few successfully decided to quit or managed to bring their habits back to a level that suited them, as shown in the following excerpt:


*The practice gradually increased and intensified in November [2020] to reach its peak. At the beginning of December, thanks to the self-banning offered on the online sites, the gambling stopped. In my case, the self-banning was more than enough to stop me (until when?). I don’t consider myself a gambler. It was a real effect of the pandemic, largely because of the huge amount of time available, and somewhat because of a secure government income [COVID-19 Canada Emergency Response Benefit].*
(Participant 350, M, age 32, PGSI score 6)

Interestingly, some new gamblers and casual gamblers before the pandemic also considered that their practices increased during the pandemic. Among them, some considered this increase moderate, but others considered it problematic.


*I used to play twice a year at the casino. One night after seeing a Loto-Quebec commercial online, I tried it. Because I had nothing to do, and I found it fun, I started playing online.*
(Participant 817, F, age 52, PGSI score 2)


*I didn’t play before the pandemic, and I had a financial cushion. Now I gamble, and I don’t have a cushion! I have developed an addiction to gambling because of confinement!*
(Participant 274, F, age 44, PGSI score 4)

Lastly, there were a few players who considered that their gambling practices had increased but that it was a source of income.

#### 3.1.2. Perceived Decrease in Gambling Practices

Some participants felt that the pandemic had led to a decrease in their gambling practices (n = 62). This group included individuals who saw gambling as an opportunity to go out, for whom the social aspect or the fact of going out in a special environment (e.g., casino) seemed to be central to their appreciation of gambling. Some of them also said that they tried online gambling but did not enjoy it much or at all. Several of these participants expressed missing their pre-pandemic gambling practices, with one going so far as to watch online slot machine videos to ease his discomfort. Among those who felt their gambling had decreased, some gamblers saw the decrease in accessibility of gambling as positive, as it helped them decrease their usual practice and they did not shift to online gambling. Many of them even believed that the pandemic was helping cure them of their gambling problems.


*The closing of the video lottery terminals in bars has made my life much better in the past year. I hope to be able to be even stronger when Loto-Quebec reopens its bar machines.*
(Participant 487, F, age 54, PGSI score 4)


*The closing of the casinos has helped me a lot. I’m more of a poker player and I don’t have as much fun playing online. Plus, online I find it much easier to leave the table than to leave the casino. For me the pandemic has helped me a lot with my gambling problem.*
(Participant 714, F, age 33, PGSI score 11)

#### 3.1.3. Perceived Stability in Gambling Practices

Beyond those participants who felt that their gambling increased or decreased, many felt the pandemic did not change their gambling (n = 104), as the following excerpt illustrates:


*Gambling has not had a negative or positive impact on my life! My habits have also not changed because I have worked as much as in the previous years. However, only the time not going out has changed. New habits have been established. More reading, board games with my family.*
(Participant 144, F, age 32, PGSI score 0)

### 3.2. Perceived Impacts of Change in Gambling Practices

Beyond describing the changes that the pandemic has had on their gambling practices, some participants expressed how these changes affected their lives. These impacts on the participants’ lives were categorized into two sub-themes: emotional impacts and financial impacts.

#### 3.2.1. Emotional Impacts

The emotional impacts of pandemic-related changes in practice (n = 134) were diverse. Those who stood out most strongly were the pleasure felt and the entertainment provided by online gambling in the context of a lack of alternative leisure. The source of social interaction that gambling represents in this context of isolation and loneliness was also experienced very positively. Other participants were relieved by the reduction in their gambling habits and sometimes took advantage of it to develop new hobbies. However, a substantial proportion of the participants experienced painful impacts from their increased gambling. These impacts ranged from depression to shame/guilt, to anxiety and regret to obsession and depression.


*I’m beyond destroyed… I’m even going to lose my home. I’m gambling all the time, hoping to make it back.*
(Participant 453, F, age 38, PGSI score 24)

#### 3.2.2. Financial Impacts

Some participants mentioned the financial impacts of changes in their gambling practices during the pandemic (n = 63). For many, increased gambling did not cause financial problems but still disheartened them.


*I started playing online casinos. Before the pandemic, I used to go to the casino once a year; now, I play every week. It doesn’t cause me financial stress, though, but I am disappointed to lose money.*
(Participant 754, F, age 35, PGSI score 5)

Others, however, clearly expressed that their pandemic gambling practices caused them major problems. For example, some people were in danger of losing their homes or gambled tens of thousands of dollars in the last year. Others went into debt or had to ask their spouses to pay the bills. Some were regular gamblers pre-pandemic, but others had their first experience with gambling during the pandemic or returned to gambling after a period of abstinence. At the other end of the spectrum were participants who explained that the reduction in their gambling practices allowed them to save money or reduce their debt, which they were proud of and found motivating in maintaining a lower level of gambling.

### 3.3. Perceived Factors Influencing Gambling Practices during the Pandemic

The factors that participants perceived to have influenced their gambling practices during the pandemic were divided into eight sub-themes. The first five reflect the impacts of the pandemic on participants’ lives, which for many resulted in a change in gambling practices: boredom and time availability, seeking positive emotions, escape, or relief from negative emotions, availability of money, and the hope of winning. The other three sub-themes focus on the impacts of the pandemic on the supply of gambling: access to gambling, gambling incentives, and gamblers’ relationship to online gambling.

#### 3.3.1. Boredom and Time Availability

The pandemic drastically changed the lives of the participants on many levels, and some of the changes were perceived to have contributed to changing their gambling practices. At the forefront was the great boredom experienced daily since the beginning of the pandemic, as well as a large amount of available time (n = 192). Participants explained that they had more free time than before the pandemic, mainly due to the impossibility of practicing their usual hobbies and the reduction of their working hours. Many people were, therefore, deeply bored and were consequently looking for ways to occupy their time. In this context, gambling appeared to be a solution for many.


*We have a lot more time to waste because we can’t socialize or have people over, so gambling is a new source of entertainment for us. We would never have played slots on a Friday night before. We used to socialize or have people over…*
(Participant 253, F, age 40, PGSI score 1)

#### 3.3.2. Search for Positive Emotions

The pandemic also deprived participants of many kinds of stimulation. In this context, gambling was seen as a source of positive emotions, fun, entertainment, and an opportunity to socialize during the pandemic (n = 46). For example, some organized and regularly played online poker games with friends and family. These games were well appreciated for the recreation they provided during times of confinement.


*I occasionally play poker with friends. We created a group during lockdown that allowed us to play poker while chatting on zoom. We played every week because it was one of the few activities we could do, and it allowed us to chat online.*
(Participant 183, M, age 41, PGSI score 0)

For others, it was the excitement and thrill that were missing and were found in the adrenaline rush provoked by gambling.

#### 3.3.3. Escape or Appeasement of Unpleasant Emotions

The pandemic, which disrupted the lives of the participants, caused a great deal of distress. Some players used gambling to escape or calm the suffering, loneliness, emptiness, or anxiety they felt during the pandemic (n = 21).


*I feel like I’ve lost everything I had built up before the pandemic, both professionally and personally. I feel like I have nothing left. Gambling simply helps me not think about it.*
(Participant 1048, F, age 29, PGSI score 11)

#### 3.3.4. Money Availability

Another aspect of the participants’ lives that the pandemic affected was the availability of money (n = 30). Many participants explained that the pandemic left them with more disposable income than usual, given the reduction in work and leisure expenses brought on by the lockdowns. Consequently, these participants felt comfortable increasing their gambling practices, considering that the money spent gambling replaces pre-pandemic leisure expenses.


*We used to go to restaurants, movies, outings. Now, with the [public health] restrictions, we take the same amount to play scratchers.*
(Participant 1451, F, age 57, PGSI score 3)

Some participants also had pay raises or were working overtime due to the pandemic and were therefore also comfortable spending more money on gambling. Others, by contrast, saw their income decreased during the pandemic, which led them to decrease their gambling practices. However, a few individuals mentioned that their decreased income caused them to gamble more because it gave them a sense of making money, or because they dreamt that it would allow them to achieve some financial security.

#### 3.3.5. Hope of Winning

Lastly, the pandemic highlighted certain life situations that contributed to the desire to gamble for some participants. They said they had been gambling more since the pandemic began in the hope of winning a good amount of money and thus being able to fulfill a dream (n = 29). For example, leaving a spouse, providing financial security for the family, and retiring earlier than expected.

#### 3.3.6. Gambling Accessibility

The space left by the pandemic in the lives of participants, combined with the accessibility of gambling (n = 69), helped explain the changes in gambling practices among participants. On the one hand, the lack of access to gambling in the presence contributed to curbing the practices of some. On the other hand, the wide accessibility of online gambling, which is available anywhere, at any time, was seen by others as a factor that contributed to the increase in their gambling practices during the pandemic.


*Also, at the beginning of the pandemic, I installed the Loto-Quebec app on my phone because we couldn’t make purchases at the convenience store. I think it increased my gambling frequency, I don’t have to go anywhere. I can pay my lottery ticket from my living room, and I make my deposits directly on the app.*
(Participant 359, M, age 30, PGSI score 1)

The fact that online gambling remained accessible during lockdowns, unlike many other leisure activities, was also perceived by many participants as contributing to the increase in practices.

#### 3.3.7. Gambling Incentives

Perceived incentives to gamble (n = 23), combined with great accessibility to online gambling, were seen by some as a reason for their increased gambling. Several participants mentioned that they had been flooded with online gambling advertisements during the pandemic, and that it made it very difficult for them to control their gambling behaviours.


*The impact: [it] has created depression. Loss of employment doesn’t help. The worst thing is that every two minutes on TV, there is an ad for online gambling, which doesn’t help the cause for a gambler.*
(Participant 258, M, age 40, PGSI score 18)

One participant also mentioned that the maximum betting limits were increased for certain games during the pandemic, which he considered to be another form of incentive.

#### 3.3.8. Participant’s Relationship to Online Gambling

Players’ relationship to online gambling (n = 109) helped explain participants’ practice changes during the pandemic. First, some gamblers reported that they had no interest in online gambling and quickly moved away from them, for example, because they did not find the social enjoyment of off-line gambling. The gambling practices of these players often decreased during the pandemic.


*It is more enjoyable to play in a place like a casino, because there is an atmosphere conducive to partying and having a good time. Playing online doesn’t offer anything close to the enjoyable casino experience.*
(Participant 235, M, age 42, PGSI score 0)

Among those who, on the contrary, said they enjoyed online gambling, we found participants for whom the pandemic led to the discovery of an online gambling offer. For some, this discovery was enjoyable and entertaining and did not lead to overindulgence, while for others, it led to major problems and was experienced as a disaster.


*I discovered the Loto-Quebec online games site to buy a Loto max [lottery ticket], and I saw that there were lots of games, and it entertains me.*
(Participant 141, M, age 41, PGSI score 4)


*Discovering online gambling caused me to lose $100,000 [CAD] in one year. I had never played online before… Anyway, hell…*
(Participant 999, M, age 47, PGSI score 25)

Further, some participants found it easier to control themselves online than during offline gambling. They benefited from the responsible gambling tools available online to help control their gambling (e.g., daily gambling limits, self-banning), and often reported that the increase in their gambling during the pandemic remained at an acceptable level to them.


*Closing casinos and gambling parlors has helped me greatly in controlling my addiction. The ability to set limits online has allowed me to put a $60-a-week marker and take more time for other things.*
(Participant 918, M, age 55, PGSI score 14)

Other participants, however, felt that it was easier to gamble excessively online than with off-line gambling. The accessibility of online games contributed to this, as did the virtual aspect of money, which led to spending without feeling that one was doing so. Some responsible gambling tools, such as self-imposed limits, did not work well for some participants who could not resist raising the limits as soon as it becomes possible again. Lastly, some mentioned that money was lost faster in online gambling and that more money could be taken out faster (e.g., from a credit card or e-check), which affected their ability to self-monitor.


*Before the pandemic, I used to go to the casino once every two weeks, but since the casino closed, I play online. And online, there is nothing to stop me from spending, whereas when I went to the casino if I wasn’t there physically, I couldn’t spend…*
(Participant 755, M, age 30, PGSI score 14)

## 4. Discussion

The objective of this article was to depict the impact of the COVID-19 pandemic on gamblers’ practices based on the answers to an open-ended question inserted into a cross-sectional survey. Such qualitative analysis of responses to an open-ended question asked as part of a cross-sectional survey is useful to contextualize questionnaire responses, to draw out aspects of participants’ experiences that could not be identified from responses to closed-ended questions, as well as to obtain results that reflect participants’ priorities and perspectives rather than those of researchers. Such research is also useful to obtain a large amount of qualitative data about the experiences of a wide range of individuals [43,44].

We found that many participants felt that their gambling practices had increased. Many explained how the pandemic created space in their lives by freeing up time and money for gambling. Some gamblers saw the pandemic as having had very positive impacts in helping them deal with their gambling problem. However, for others, the pandemic had devastating effects. Some felt that they worsened their gambling problems considerably, while others reported that they had developed a gambling problem, which they did not have before the pandemic. These results led to the following observations.

We found that a substantial proportion of gamblers felt that their gambling had increased during the pandemic. This result should be interpreted with caution, although it is consistent with the concerns expressed by many experts about an increase in gambling problems during the pandemic [6]. It has been demonstrated that self-reported gambling spending does not necessarily represent actual spending [47]. This might be especially true in the context of the pandemic, which by disrupting all participants’ gambling habits might have made it more difficult to estimate gambling spending. However, regular gamblers generally tend to underestimate their gambling expenditures rather than overestimate them [47], which suggests that the perceived increase expressed by this study’s participants may be justified, or even underestimated. It is also important to remember that 57% of the sample in this study was made up of current problem gamblers, while studies have shown that they represent between 1% and 3% of the general population [13,14] and between 5% and 40% of online gamblers [31,32,33]. Moreover, while the empirical results reported so far show a decrease in gambling practices during the first weeks of the pandemic [4,18,48,49], the few increases observed were consistently associated with problem gamblers [18,49,50,51]. Thus, the results of this study are interesting, as they reflect the experience of problem gamblers, who are particularly at risk, especially during a global crisis such as the pandemic, but they remain not generalizable.

We also noted that many gamblers who reported an increase in their gambling practices said that it was not having a deleterious effect on them or their lives. For example, some said that the increase in gambling was not a problem because they had more money available due to the restrictions that the health measures had placed on their leisure or work habits. The pandemic made room in their lives for more gambling. By contrast, others expressed great distress at the increase in their gambling habits. Thus, variations in gambling practices during the pandemic must be interpreted with caution, as they may underlie a variety of realities [52] that qualitative studies can help shed light on.

It is also important to note that some gamblers benefited greatly from COVID-19 public health measures in that they were able to considerably reduce their problem of gambling [48], which was experienced by some as life-saving. The pandemic thus represented a unique window of opportunity for some gamblers to change their habits [53].

This study has limitations. First, the length of responses varied greatly between participants, some of them being very limited. Second, the data collection method made it impossible to question participants further than what first came to their minds. This impossibility makes it difficult to assess the scope of the identified themes because not mentioning a theme does not imply its exclusion from the participant’s experience.

## 5. Conclusions

This study sheds light on the experiences of gamblers, especially problem gamblers, during the COVID-19 pandemic. We found that some gamblers were at great risk with the pandemic. Documenting their experiences is important, as we know that the impact of problem gambling can go beyond the individual to the family and community [14]. It also showed that variations in gambling practices must be interpreted with caution, as they may underlie a variety of realities. Finally, this study provided qualitative access to the experiences of many participants while revealing aspects of their experiences of the highest relevance to them. As such, this study presents the essence of what these participants felt was important to express about their gambling experiences during the COVID-19 pandemic.

Future research using semi-structured interviews will deepen the themes raised in this study, for example, the relationship between participants’ relationship to online gambling and the direction (increase, decrease, or stability) of the changes in their practices during the pandemic. It will also look at the experiences of those close to the gamblers, who are directly affected by and sometimes witness the gambling practices of the participants.

## Figures and Tables

**Table 1 ijerph-19-16603-t001:** Sociodemographic portrait of participants.

	Respondents (n = 724)
Sex	
• Female	389 (54%)
Age (years)—med (EIQ) ^a^	43 (33–55)
Education	
• No diploma	37 (5%)
• High school	250 (35%)
• College	178 (25%)
• University	254 (35%)
Occupation	
• Employee	420 (58%)
• Independent worker	69 (10%)
• Unemployed	49 (7%)
• Student	60 (8%)
• Retiree	93 (13%)
• Other	31 (4%)
PGSI	
• Non-problem gamblers	171 (24%)
• Low-risk gamblers	129 (19%)
• Moderate-risks gamblers	210 (28%)
• High-risk gamblers	214 (29%)

^a^ median (interquartile range).

**Table 2 ijerph-19-16603-t002:** Thematic framework of the experience of gamblers during the COVID-19 pandemic.

Theme	Sub-Theme
Perceived changes in gambling practices during the pandemic	Perceived increase in gambling practices (n = 238) Perceived decrease in gambling practices (n = 62) Perceived stability of gambling practices (n = 104)
Perceived impacts of changes in gambling practices	Emotional impacts (n = 134) Financial impacts (n = 63)
Perceived factors influencing gambling practices during the pandemic	Boredom and time availability (n = 192) Search for positive emotions (n = 46) Escape or appeasement of unpleasant emotions (n = 21) Money availability (n = 30) Hope of winning (n = 29) Gambling accessibility (n = 69) Gambling incentives (n = 23) Participants’ relationship to online gambling (n = 109)

## Data Availability

All data, including original texts in French, is available from the corresponding author on reasonable request.

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
