# Peer review of "Gamblers’ Perceptions of the Impact of the COVID-19 Pandemic on Their Gambling Behaviours: Analysis of Free-Text Responses Collected through a Cross-Sectional Online Survey"

_ijerph, 2022, doi:10.3390/ijerph192416603_

Round 1
Reviewer 1 Report
I thank the authors for the opportunity to review this manuscript, which I find very interesting but needs some improvements:
1) Abstract is not clear that the participants first answered a questionnaire with which it was possible to divide the participants
2) Introduction is very poor, I suggest expanding it in light of the present literature on the topic and make it clear what the research hypotheses are.
3) In the method section, it seems to me that there is no logic for a person who is not familiar with research. It is appropriate to explain in stages what instruments were analyzed and what types of analysis were conducted, both for the quantitative and qualitative parts.
4) How were the reading categories constructed?
5) It would be interesting to use a cloud analysis to see what were the most recurrent words in the research (eg. See "Forced Cohabitation during Coronavirus Lockdown in Italy: A Study on Coping, Stress and Emotions among Different Family Patterns" Mari et al 2020)
6) Why in the results did you not report the analysis in the different groups of player types? It might be interesting to see what differences there are according to severity.
Best Regards
Author Response
Response to Reviewer 1 Comments
Point 1: Abstract is not clear that the participants first answered a questionnaire with which it was possible to divide the participants.
Response 1: Thank you for this comment. We have reviewed our abstract and added precisions to make this point clearer.
Point 2: Introduction is very poor, I suggest expanding it in light of the present literature on the topic and make it clear what the research hypotheses are.
Response 2: The introduction has been modified accordingly. We have added recent literature as suggested and we hope our objective is also easier to understand. Our work is not based on a hypothetico-deductive model but rather on an inductive methodology, based on a research objective. This explains why no hypothesis was formulated beforehand, as we followed a classical inductive methodological approach.
Point 3: In the method section, it seems to me that there is no logic for a person who is not familiar with research. It is appropriate to explain in stages what instruments were analyzed and what types of analysis were conducted, both for the quantitative and qualitative parts.
Response 3: Thank you for your comment. We have clarified this (see section 2.5 Analysis). We hope this helps.
Point 4: How were the reading categories constructed?
Response 4: As mentioned above, we followed a classical model of qualitative thematic analysis. This implies that the categories were constructed inductively and iteratively, using the NVivo software for coding and organizing all emerging themes during the analysis process. We have made this more explicit in the article (section 2.5 Analysis).
Point 5: It would be interesting to use a cloud analysis to see what were the most recurrent words in the research (eg. See "Forced Cohabitation during Coronavirus Lockdown in Italy: A Study on Coping, Stress and Emotions among Different Family Patterns" Mari et al 2020)
Response 5: Thank you for this interesting suggestion. We will keep this in mind for future publications. In this case, we rather opted for an inductive thematic analysis, using NVivo, as our aim was to study qualitatively the emerging/recurring themes mentioned by participants, rather than the specific words or expressions chosen to express these ideas or experiences. But discourse analysis is also fascinating, and we might do this in the future and include cloud analysis.
Point 6: Why in the results did you not report the analysis in the different groups of player types? It might be interesting to see what differences there are according to severity.
Response 6: Thank you for this suggestion. We agree this is interesting. In fact, we are currently working on a qualitative analysis article of all survey data, which will include transversal links between gambling severity scores and many other variables. For this article, we decided to adopt a classic qualitative analysis as the question was open-ended and our methodology qualitative. Therefore, scores were not linked with themes. But we have now added the severity PGSI scores of participants, as well as gender and age, after each direct citation. We think this adds interesting information to the reader, even If it is not generalizable to all answers/participants being associated with a theme.
Reviewer 2 Report
The article overall is a well written and well laid out. However there are some areas of potential improvement.
Materials and Methods
Study design - There is reference to the second stage of the study which is not reported in this article. This statement does not belong in the methodology section for this article, it would be more appropriate to include as part of the discussion as future research.
Analysis - As the data was collected in French, it would be inferred that the analysis was undertaken in French and then translated for the purposes of reporting. It would be ideal to explicitly state this. It may also be useful to have some of the original French text included with the quotes to ensure nothing gets lost in translation.
Results
- A minor typo "969 completed the questionnaire at to the end"
- With regards to the table 1 characteristics of participants my quantitative brain is interested in how the characteristics of those that completed the free text question differed from the rest of the survey participants as there are often major differences here. However this is not critical to the qualitative analysis.
- In table 2, I am not sure of the need of the example responses in this table since each of the sub-themes is discussed in the following sections with example responses, and feels slightly repetitive.
Results and Discussion
- there is a statement of there being a "significant proportion" of participants/gamblers a number of times in the results and discussion, which is carried through to the abstract. Given the fact that "significant" is term that is used in quantitative research indicating that a statistical test has been undertaken, I recommend that an alternative term is used.
- The reported results and discussion have just focused on the free text content without incorporating any of the participants characteristics. This is especially critical with PGSI status, but potentially relevant with some of the other characteristics. As increases in gambling is a particular concern for problem or at risk gamblers as this may have more financial or mental health impacts. Similarly decrease in gambling for problem or at risk gamblers may lead to positive outcomes. A bit more in depth examination of the themes with these characteristics and related dynamics would strengthen this article significantly.
Author Response
Response to Reviewer 2 Comments
Point 1: Study design - There is reference to the second stage of the study which is not reported in this article. This statement does not belong in the methodology section for this article, it would be more appropriate to include as part of the discussion as future research.
Response 1: Thank you. We have moved this to the discussion section as suggested.
Point 2: Analysis - As the data was collected in French, it would be inferred that the analysis was undertaken in French and then translated for the purposes of reporting. It would be ideal to explicitly state this. It may also be useful to have some of the original French text included with the quotes to ensure nothing gets lost in translation.
Response 2: You are right, the survey was conducted in French, as well as all participants’ answers. The analysis and writing of this article were also completed in French before being translated in English. The English version of the article was revised linguistically by a professional and this revision verified by all coauthors (who are all fluent in English and able to verify the translation for any misinterpretations). We are aware of the limits of translations, and you bring up a good point. To keep the article simple, we prefer not to include the original French citations along the translated text. But all original data is of course available to those interested upon request to the corresponding author. We have added a note mentioning this at the end of the article.
Point 3: A minor typo "969 completed the questionnaire at to the end"
Response 3: This typo has been corrected, thank you.
Point 4: With regards to the table 1 characteristics of participants my quantitative brain is interested in how the characteristics of those that completed the free text question differed from the rest of the survey participants as there are often major differences here. However, this is not critical to the qualitative analysis.
Response 4: 969 participants completed the online survey. A total of 721 of them decided to answer the optional open question at the end (81st question), which invited them to express in their own words the impact of the COVID-19 pandemic on their personal gambling practices. The present article is qualitative in nature, which explains why more detailed quantitative data has not been added to this article. We are currently working on an in-depth quantitative article from the analysis of all data from the survey and this will hopefully answer your curiosity! We hope this will be published early 2023.
Point 5: In table 2, I am not sure of the need of the example responses in this table since each of the sub-themes is discussed in the following sections with example responses, and feels slightly repetitive.
Response 5: Thank you, that is a good point. We have removed the examples in Table 2.
Point 6: Results and Discussion - there is a statement of there being a "significant proportion" of participants/gamblers a number of times in the results and discussion, which is carried through to the abstract. Given the fact that "significant" is term that is used in quantitative research indicating that a statistical test has been undertaken, I recommend that an alternative term is used.
Response 6: Thank you, this has been modified and linguistic alternatives found to show recurrence.
Point 7: The reported results and discussion have just focused on the free text content without incorporating any of the participants characteristics. This is especially critical with PGSI status, but potentially relevant with some of the other characteristics. As increases in gambling is a particular concern for problem or at risk gamblers as this may have more financial or mental health impacts. Similarly decrease in gambling for problem or at risk gamblers may lead to positive outcomes. A bit more in depth examination of the themes with these characteristics and related dynamics would strengthen this article significantly.
Response 7: Thank you for this suggestion. As mentioned above, we agree this is interesting. In fact, we are currently working on a qualitative analysis article of all survey data, which will include transversal links between gambling severity scores and many other variables. For this article, we decided to adopt a more classic qualitative analysis as the question was open-ended and our methodology qualitative. Therefore, scores were not linked with themes. But we have now added the severity PGSI scores of participants, as well as gender and age, after each direct citation. We think this adds interesting information to the reader, even If it is not generalizable to all answers/participants being associated with a theme.
Round 2
Reviewer 1 Report
The manuscript has been edited, but the introduction still seems poor; I suggest extending it on the topics addressed in the manuscript.
Author Response
Point 1: The manuscript has been edited, but the introduction still seems poor; I suggest extending it on the topics addressed in the manuscript.
Response 1: Thank you for this comment. We did a second round of editing to improve the introduction. We added 6 references (for a total of 22 new references since the first round of revision). A new paragraph on the impact of the Covid-19 pandemic on gambling behaviors (the main theme of our study) was also added to improve this section.
Reviewer 2 Report
Two minor points:
Table 2 still has the formatting for the extra column and could be tidied up and take up less space.
In section 2.5 analysis section I believe a sentence to state that the analysis was carried out in french and the results professionally translated for the purpose of reporting. it is best to explicitly state this although it could be inferred.
Author Response
Point 1: Table 2 still has the formatting for the extra column and could be tidied up and take up less space.
Response 1: We have modified the formatting of Table 2, as requested.
Point 2: In section 2.5 analysis section I believe a sentence to state that the analysis was carried out in french and the results professionally translated for the purpose of reporting. it is best to explicitly state this although it could be inferred.
Response 2: Thank you for this comment. We have added a sentence in section 2.5 to make this explicit.